psychology

tainted altruism, direct replication, pre-registered study

**Author for correspondence:**
Harold Pashler
e-mail: hpashler@gmail.com

# The tainted altruism effect: a successful pre-registered replication

Valerie Alcala, Kendra Johnson, Caroline Steele, Juanshu Wu, Donglai Zhang and Harold Pashler

Department of Psychology, University of California, San Diego, 9300 Gilman Drive, La Jolla, CA 92093, USA

HP, 0000-0003-3644-4909

Newman and Cain (Newman, Cain 2014 *Psychol. Sci.* **25**, 648–655 (doi:10.1177/0956797613504785)) reported that observers view a person's choices as less ethical when that person has acted in response to both altruistic and selfish (commercial) motivations, as compared with purely selfish interests. The altruistic component reduces the observers' approval rather than raising it. This puzzling phenomenon termed the 'tainted altruism' effect, has attracted considerable interest but no direct replications in prior research. We report direct replications of Newman and Cain's Experiments 2 and 3, using a larger sample ($n = 501$) intended to be fairly representative of the US population. The results confirm the original findings in considerable detail.

## 1. Tainted altruism: a successful pre-registered replication

People frequently choose behaviours that fulfil different kinds of motivations. For example, they often seek both selfish and altruistic goals at the same time. How are these mixed-motivated behaviours perceived by others? This question has sparked much discussion, especially among social psychologists.

Newman & Cain [1] described a phenomenon they termed 'tainted altruism'—when an ostensibly prosocial action also serves some private goal, the action is viewed as less virtuous than a purely selfish act. Tainted altruism affects not only an observers' moral appraisals of another person's action and characters but also their own behavioural choices. This influential paper has been cited over 150 times (Google Scholar, 14 June 2021) and is part of a broader literature examining the reactions that others have to prosocial behaviour (for two very recent reviews, see [2], and [3]).

Substantially cited papers need and deserve independent replication. As the famous Nosek *et al.* [4] reproducibility project

(and the results of many other recent replication attempts) indicate, only a modest fraction of published psychological research findings can be replicated. Many researchers seem to have been genuinely surprised by these failures, even though statisticians had been cautioning about the likelihood of high unreproducibility for decades. Obviously, it is important for any field to identify which important findings are replicable, and which are not, and the general neglect of this step is sure to impede progress (cf. [5]).

Our project aimed to replicate two of Newman and Cain's studies which seem to us to well embody these authors' fundamental claims regarding the tainted altruism effect. We used a considerably larger sample than the original article and the study was pre-registered to pin down our research hypothesis, methods and analysis plans before the data collection commenced. Although pre-registration is more common now than in the past, this potentially beneficial research practice appears absent from studies on this broad topic [6]. We chose to replicate Newman and Cain's Experiments 3 and 2 (which will be presented in that order) because we felt that this sequence embodied the key findings of the research in the simplest and clearest way among all the available choices.

## 1.1. Current research

Our first replication aimed to verify Newman and Cain's Experiment 3 finding. In this study, participants were asked to morally appraise a hypothetical person's action and character given various factors. Our second replication aimed to verify Newman and Cain's Experiment 2 finding, which featured a more quantitative measurement of the tainted altruism effect. Our replication was pre-registered on AsPredicted.com where it can be accessed through this link: https://aspredicted.org/~7E94fcbelU.

# 2. Experiment 1: direct replication of tainted altruism effect

In this experiment, participants were randomly assigned to one of four groups. In the first and second groups, the participants read a short passage in which a business owner either donated money to charity (Charity condition) or invested the money in advertising (Advertising condition) with the goal of increasing revenues for his business. All passages were taken directly from the supplemental materials of the original study. The passage that participants in the Charity condition read was:

> Frank Mulberry is the owner of a large chain of 'Mulberry's' department stores located throughout the Midwest. Recently, Mulberry donated several million dollars to a local children's hospital in Omaha, Nebraska, where his company is based. Mulberry donated the money because he knew that the good publicity would boost the reputation of his company and get more people to come to his stores.

In the Advertising condition, the passage read:

> Frank Mulberry is the owner of a large chain of 'Mulberry's' department stores located throughout the Midwest. Recently, Mulberry invested several million dollars in an extensive advertising campaign in Omaha, Nebraska, where his company is based. Mulberry invested the money because he knew that the good publicity would boost the reputation of his company and get more people to come to his stores.

Participants then indicated their evaluation of Mulberry on several different dimensions. The basic tainted altruism effect consisted in people offering a *lower* evaluation of Mulberry in the Charity condition as compared with the Advertising condition. Indeed, in the original study, Mulberry lost, rather than gained, in the participant' estimation for having decided to direct his money in a way that served charitable *plus* self-interested purposes as compared with having made a purely self-interested purchase of some advertising with no charitable aspect.

The original study included two additional conditions intended to help test the authors' hypotheses about the cause of the tainted altruism effect, and we included these conditions in our replication. Specifically, the authors surmised that the basic tainted altruism effect might be nullified or reversed if language were added to the vignettes that would lead participants to contrast Mulberry with a hypothetical oppositely motivated decision maker. This yielded a third and fourth condition. In the Advertising with Counterfactual condition subjects read the following scenario (the added language is shown in italics):

> Frank Mulberry is the owner of a large chain of 'Mulberry's' department stores located throughout the Midwest. Recently, Mulberry invested several million dollars in an extensive advertising campaign in Omaha, Nebraska, where his company is based. Mulberry invested the money because he knew that the good publicity would boost the reputation of his company and get more people to come to his stores. *Keep in mind that if he wanted*

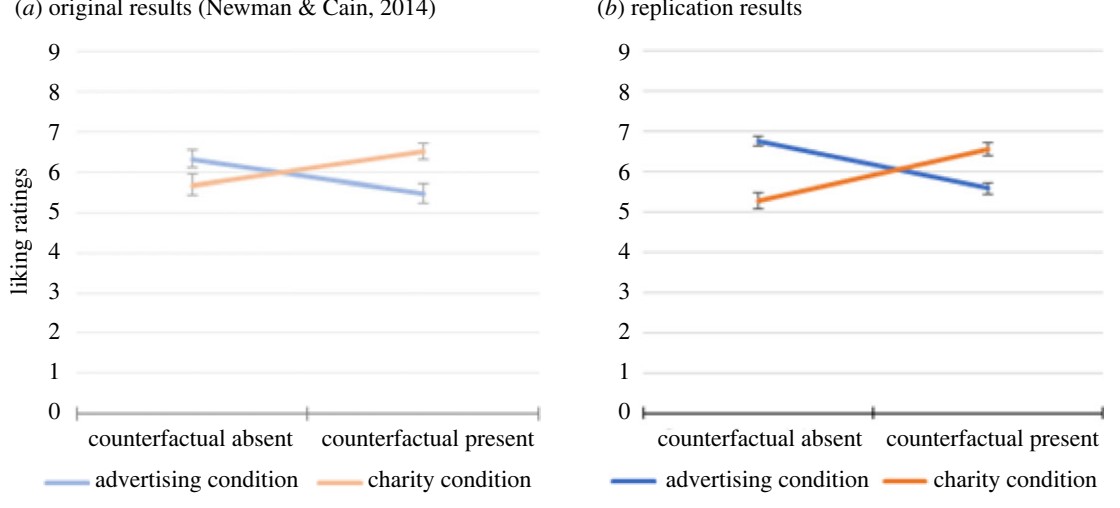

**Figure 1.** Liking ratings in the original study (Panel *a*) and the current replication study (Panel *b*) as a function of condition (Advertising versus Charity) and absence/presence of the counterfactual information. Error bars represent standard error of the mean.

*to, Mulberry could have instead donated the money to charity. This would have also increased the reputation of his company, but all of the money would have gone to charity.*

In the Charity with Counterfactual condition, subjects read (again, the added language is shown in italics):

Frank Mulberry is the owner of a large chain of 'Mulberry's' department stores located throughout the Midwest. Recently, Mulberry donated several million dollars to a local children's hospital in Omaha, Nebraska, where his company is based. Mulberry donated the money because he knew that the good publicity would boost the reputation of his company and get more people to come to his stores. *Keep in mind that if he wanted to, Mulberry could have instead invested the money in advertising. This would have also increased the reputation of his company, but none of the money would have gone to charity.*

In the original study, adding the counterfactual language reversed the tainted altruism effect (figure 1*a*).

## 2.1. Method

### 2.1.1. Participants

Five hundred and one participants (mean age = 46.3 years, 51% female, 49% male) were recruited through Prolific.org, providing us with a representative sample of people residing in the US with regard to age, sex and ethnicity [7]. During the subject recruiting process, Prolific used three pre-screeners: 'Date of Birth', 'Sex', and 'Ethnicity (Simplified)' to stratify age using five 9-year brackets (18–27, 28–37, 38–47, 48–57 and 58+), sex using male and female category and Ethnicity using five categories (White, Mixed, Asian, Black and Other). Six subjects failed an 'attention check' by neglecting to follow an explicit instruction about how to respond to an item included to assess attentiveness.

### 2.1.2. Design and procedure

The design and procedure followed the target study by Newman & Cain [1] in all particulars except as noted.

In all conditions, participants rated Mulberry's behaviour on various dimensions using a 9-point scale (1 = *absolutely not* to 9 = *absolutely*). Participants were asked the following questions: 'How ethical is Mulberry?', 'How much do you approve of Mulberry's behaviour?', 'How much do you like Mulberry?', and 'Did Mulberry act as altruistically as he could have?'.

One additional question (not included in the original study) was asked in our experiment on an exploratory basis: 'How deceptive do you think Mulberry is?' (our reasons for including this are described below).

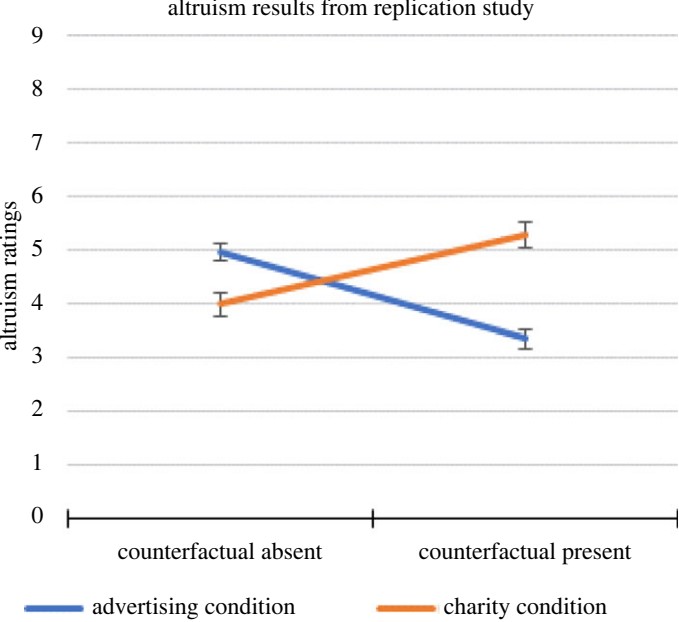

**Figure 2.** Altruism ratings in the current replication study as a function of condition (Advertising versus Charity) and absence/presence of the counterfactual information. Error bars represent standard error of the mean.

## 2.2. Results and discussion

The complete dataset for this study is public at https://tinyurl.com/2bvkwr9z.

The analysis code for this study is public at https://figshare.com/s/f67f46b85cf9a756be7a.

As was done in the original experiment, each subject's responses to the first three questions were averaged together to yield an overall liking rating. (The Cronbach's alpha for these three liking questions was 0.899, suggesting high statistical coherence.) As shown in figure 1b, the average values for the overall liking rating were as follows: Charity = 5.28 (s.e. = 0.19), Advertising = 6.76 (s.e. = 0.12), Charity with Counterfactual = 6.56 (s.e. = 0.17), Advertising with Counterfactual = 5.58 (s.e. = 0.16). We defined the difference between the averaged liking ratings from the Charity and Advertising conditions (without counterfactual information) as the basic effect of tainted altruism. An unusually large effect size of Cohen's d = 0.84 was observed for the effect (Charity versus Advertising without counterfactual information in either). The contrast of the two conditions was significant, $t_{206} = 6.82$, $p < 0.001$.

However, as shown in figure 1, when the counterfactual information was provided, the effects of condition (charity versus advertising) on liking was reversed, such that participants rated the target more positively when he gave to charity than when he invested the same money in advertising, $t_{226} = -4.29$, $p < 0.001$.

A 2 × 2 analysis of variance revealed a significant interaction between domain (charity, advertising) and the presence/absence of counterfactual information, $F_{1,490} = 63.12$, $p < 0.001$. As figure 1 illustrates, the cross-over interaction found in our study closely resembled the corresponding interaction in the original study.

As to the question of how altruistically Mulberry acted, the average values were Charity = 3.98 (s.e. = 0.22), Advertising = 4.96 (s.e. = 0.18), Charity with Counterfactual = 5.28 (s.e. = 0.24) and Advertising with Counterfactual = 3.34 (s.e. = 0.19). The same pattern of cross-over interaction was evident as with the liking questions. This is shown in figure 2.

For the Deceptiveness question (added in our study on an exploratory basis) the results are shown in figure 3.

This indicates that the derogation of Mulberry in the charitable condition extends to perceptions of how deceptive he is, as well as how moral and altruistic he is. Subjects may view profiting personally from charity as sneaky and misleading, and this may explain the derogation seen in this condition. Whenever a personal gain is made from charity, the person profiting may be perceived as having sought to increase their social standing by encouraging others to misinterpret their motives. Our results confirmed the effect of the condition manipulation on perceptions of deceptiveness: Mulberry was rated significantly more deceptive in the Charity condition ($M = 5.06$) than in the Advertising

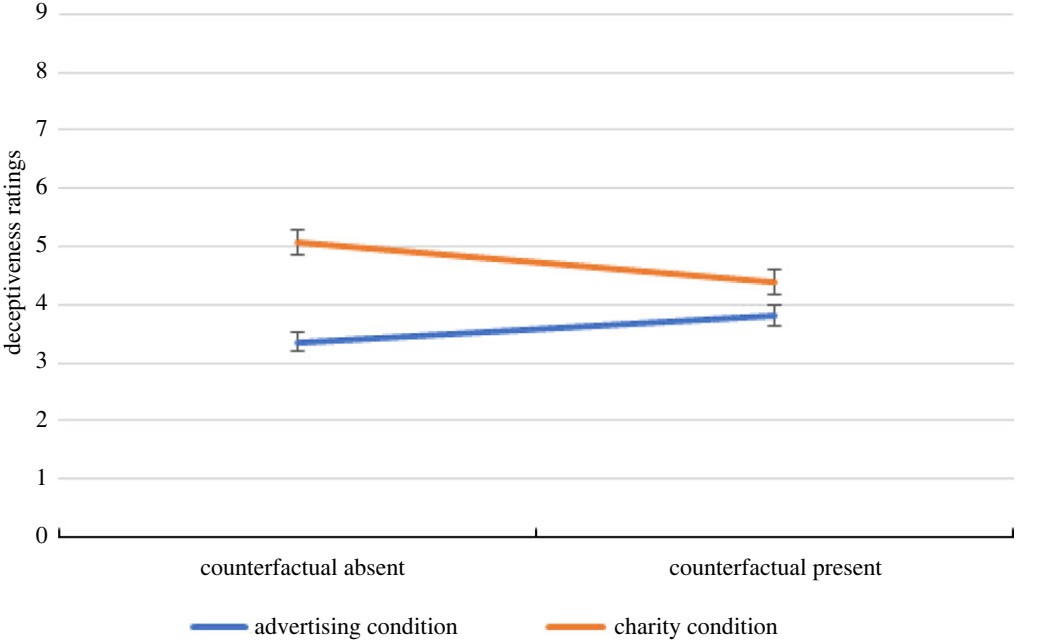

**Figure 3.** Deceptiveness ratings in the current replication study as a function of condition (Advertising versus Charity) and absence/presence of the counterfactual information. Error bars represent standard error of the mean.

condition (3.36) ($M = 5.06$), $t_{238} = 6.24$, $p < 0.001$. Furthermore, when participants are reminded that Mulberry either could have acted charitably or did not have to act charitably by the counterfactual information, Mulberry was seen as less deceptive for donating to charity ($M = 4.34$), $t_{232} = 2.22$, $p = 0.014$. Interestingly, there was no reversal of the effect, however. In sum, the results seem consistent with the idea that attributions of deceptiveness may be a key part of the mechanism underlying the effect.

# 3. Experiment 2: impact of tainted altruism effect on monetary decisions

Our second study (conducted with the same 501 participants as the first) was a replication of Experiment 2 from Newman & Cain's [1] original paper. The aim of this experiment was to determine whether the tainted-altruism effect would extend to participants' behavioural intentions as reported in the original paper. Specifically, to quote the original investigator, the design examined whether people were 'willing to forgo the opportunity to earn more money for a charity if the person raising the money also earned a substantial profit' [1].

## 3.1. Method

### 3.1.1. Participants

The same 501 participants completed the questions included in this experiment after completing the first experiment.

### 3.1.2. Design and procedure

Participants were randomly assigned to one of two between-subjects conditions labelled Charity or Corporation. Participants were asked to imagine that they were the head of either a charity or a corporation, charged with the responsibility of choosing a promoter to raise funds for their organization. They then read the following passage about a potential promoter named Daniel P. (the passage was taken directly from the supplemental materials of the original study).

> Daniel P.'s organization is a for-profit company. After all of the organizational costs have been settled, he and his staff take a percentage of the remaining funds. Therefore, the more money that is earned for the charitable cause, the more profits go to Daniel P. and his staff.

Eleven binary hiring choices were then presented to participants (Newman and Cain stated that they used 10 choices but the supplementary materials showed 11). Each choice asked them whether they

would choose to hire Daniel P.'s firm, who charged more but potentially raised more, or an alternative promoter, who charged less in fees but also raised less total money. Every choice of an alternative promoter in one of these choices can be seen as a costly and thus irrational choice presumably driven by the tainted altruism effect.

The order in which these questions were asked was fixed across subjects (see electronic supplementary material for verbatim details). All these binary choices were presented on the same web page which participants scrolled down to view (submitting the web page when they were ready). We confirmed with the original authors that this matched the procedure of the original study (Newman, 21 April 2021, personal communication). After making their hypothetical hiring decisions, participants were asked to rate Daniel P. across several measures. Participants rated how ethical he was (1 = *completely unethical*, 9 = *completely ethical*), how moral he was (1 = *completely immoral*, 9 = *completely moral*), and the extent to which they approved or disapproved of his actions (1 = *definitely not*, 9 = *definitely so*). Participants also rated how beneficial his actions were (1 = *not at all*, 9 = *very beneficial*), and the extent to which he 'made the world a better place' (1 = *not at all*, 9 = *very much so*). Exact wording of all questions is found in the electronic supplementary material.

## 3.2. Results and discussion

The complete dataset for this study is public at: https://tinyurl.com/2bvkwr9z.

The analysis code for this study is public at https://figshare.com/s/f67f46b85cf9a756be7a.

Following the original study, participants' data were excluded if either of two conditions were held. The first was if they failed an initial comprehension check question (excluding 123 subjects). The second was if their set of choices violated an order constraint specified in the original study (excluding another 26), described further below. A total of 149 subjects' data were excluded.

The comprehension check was a single binary hiring decision question presented before the eleven binary hiring questions that comprised this experiment. In this comprehension check, participants were asked to choose between Daniel P.'s firm who charged more and earned less than that of an alternative promoter which charged less and earned more. Participants who selected Daniel P.'s firm (an indefensible choice regardless of the relative importance of the two factors) were excluded from the data analysis. As in the original experiment, other participants were also excluded from data analysis if they made choices that broke a rather subtle ordering constraint. Specifically, a participant was excluded if they chose the alternative firm on question n, switched to choosing Daniel's firm on some later question $n + j$ (for some positive integer j), and then switched back to choosing the alternative firm on question $n + k$ (for some positive integer $k > j$). The original authors' rationale for doing this, we surmise, was basically that such a set of choices can be shown to be incoherent based on some reasonable assumptions including transitivity of preference.

Responses to the first three rating questions were averaged together to produce a single measure of morality and the last two rating questions were averaged to produce a single measure of perceived benefit. The Cronbach's alpha values were 0.927 for the morality questions and 0.844 for the benefit questions. Participants in the charity condition rated Daniel P. as significantly less moral ($M = 5.16$, s.e. = 0.16) than did participants in the corporate condition ($M = 5.74$, s.e. = 0.15), $t_{343} = 2.64$, $p = 0.004$. Ratings of perceived benefit did not differ significantly across the two conditions, $M = 5.56$, s.e. = 0.16 for the Charity condition and M = 5.47, s.e. = 0.15 for the Corporate condition ($t_{343} = 0.40$, $p = 0.69$).

The behavioural choice data are shown below in figure 4.

## 3.3. Exclusion of subjects

One possible concern with both the original study and our replication relates to the exclusionary criteria used prior to data analysis. The use of the comprehension check question and the order constraint on participant decision making (described above) led to a large number of participants' data being excluded. In Newman and Cain's second experiment, 23% of their participants were excluded and in our replication, 30% of participants were excluded for the same reasons. In our view, the authors' decision to exclude participants using these criteria is understandable and arguably even conservative with respect to the hypothesis being tested. However, it seemed worth checking whether these exclusionary criteria might somehow have created or exaggerated the tainted altruism effect. To assess this, we compared the contrast results obtained with exclusion (reported above) with the findings obtained when we did no data exclusion at all.

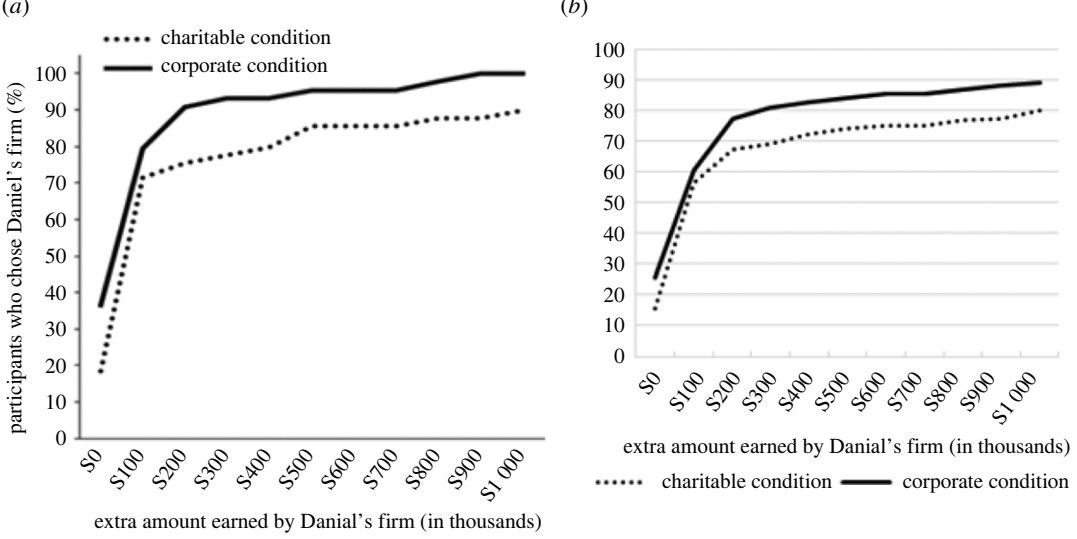

**Figure 4.** Panel (*a*) shows the results of Experiment 2 of Newman and Cain's original tainted altruism experiment, and panel (*b*) shows the results of current replication. Lines show the percentage of participants in each condition who selected Daniel's firm as a function of the extra amount Daniel's firm earned (and whether the condition was Corporate or Charity).

Without any data exclusion, for the morality measure, the means were 5.92 and 5.40 for Corporate and Charity conditions, respectively, a significant difference in the expected direction, $t_{495} = 2.89$, $p = 0.004$. This paralleled the results found with exclusion. For the benefit measure, the means were 5.72 and 5.75 for Corporate and Charity conditions, respectively. As with the results after data exclusion, this too yielded no significant difference between conditions, $t_{495} = 0.14$, $p = 0.89$.

Thus, while the data exclusion policy followed by the original authors (and replicated in our first analysis reported above) was unusually severe, it does not seem to have substantially affected the outcome of the replication study.

## 3.4. Summary

Our efforts yielded a highly successful replication of Newman & Cain's second experiment. This lends further support to the validity of the basic tainted altruism effect, with participants in the Charity condition rating Daniel as less moral than did participants in the Corporate condition. Morality ratings differed significantly between the two conditions, which indicates that profit-seeking itself was not seen as tainting. It was the presence of self-interest in the charitable domain specifically that resulted in tainted evaluations/decreased ratings of morality. Even more importantly, the tainted altruism effect extended not only to moral evaluations of others but seems to have driven participants' own behavioural intentions. As was observed by Newman & Cain [1], participants in the Charity condition were willing to forgo the opportunity to make more money for their charity than they were willing to forgo in the Corporate condition (figure 4). The match between the original and replication functions underscores the broad success of the present replication.

## 4. General discussion

The replication crisis has generated concern and worry across the field of psychology for the last decade. Too few replications have been conducted to provide a broad sense of the validity of the psychological literature as a whole, but the results to date have not been terribly encouraging. Canny observers have long noted that even in the presence of the usual bias toward publishing positive effects, there can also be a 'reverse publication bias' that might favour the publication of *failures* to replicate more than successful replications like the current paper. This can potentially distort the literature in the opposite direction from the classic publication bias effect [8–10]. Clearly, in order to see reality accurately, careful direct replications need to be conducted frequently (at least for influential findings) and published without regard to their outcome.

To the best of our knowledge, this study is the first direct replication of any of the experiments in Newman & Cain's [1] study on tainted altruism. Using a larger sample size and a US sample

intended to be fairly representative of the US population at least with respect to age, sex and ethnicity, we readily reproduced the main results in the original studies quite closely.

## 5. Concluding comments

We end with a few comments on the broader substantive issues raised by Newman & Cain's [1] work. Their study was not the first to explore the conflict between perceived altruism and potential selfishness and can be tied to some broad themes in the literature. The term 'attributional cynicism' was used by Critcher and Dunning to describe the phenomenon in which people tend to judge seemingly selfless acts as more selfish than seems warranted in terms of Bayesian reasoning [11]. Another group theorized that charitable credits are 'cheapened' because of perceived emotional selfishness when the prosocial actor has a personal connection to the charitable cause [12]. While attributional cynicism may lead people to devalue prosocial actions with non-altruistic components, Newman and Cain's study (confirmed in the present replication report) went further in showing that people also devalue a prosocial act if the intention is even partly selfish [1]. Later research has brought out a number of findings that may reflect the effect of tainted altruism. For example, bragging about good deeds may undermine attributions of generosity [13] and observers penalize self-oriented motives when judging altruistic behaviours [14].

Now that the solidity of the tainted altruism effect seems well established, future research might explore the cause and mechanism of the effect in greater detail, addressing why and how a selfish motive undermines the judgement of prosocial behaviour. The concept of a conflict between two different forms of exchange markets [15] may help shed light on this. Charitable actions belong to the social market, where people's behaviour is assumed to be guided by altruism. On the other hand, selfish motives are viewed as unobjectionable when confined to the monetary market, where behaviour is expected to be based upon reciprocity. The tension between different exchange categories may cause observers to view as inauthentic prosociality caused even in part by extrinsic motives [16]. The findings presented above regarding judgements of Deceptiveness seem notably congenial to this interpretation, suggesting that some observers readily put the worst possible interpretation on behaviours not easily allocated to the realm of one of these markets or the other.

Ethics. The research was approved by the University of California Institutional Review Board (Protocol 140207).

Data accessibility. The complete dataset for this study is public along with R code for analysis of the data at: https://tinyurl.com/2bvkwr9z. The data are provided in electronic supplementary material [17].

Authors' contributions. V.A.: conceptualization, project administration, writing—original draft, writing—review and editing; K.J.: conceptualization, project administration, writing—original draft; C.S.: conceptualization, project administration, writing—original draft; J.W.: Data curation, formal analysis, software, visualization; D.Z.: data curation, formal analysis, validation, visualization, writing—original draft, writing—review and editing; H.P.: conceptualization, investigation, methodology, supervision, writing—original draft, writing—review and editing. All authors gave final approval for publication and agreed to be held accountable for the work performed therein.

Competing interests. We declare we have no competing interest.

Funding. We received no funding for this study.

Acknowledgements. The authors are grateful to the original authors for answering questions about details of the original study. The authors are grateful to Julia M. Rohrer for her very useful comments on an earlier version of this paper.

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
