## [Peer Review File · Royal Society Open Science]

Review History

RSOS-211152.R0 (Original submission)

Review form: Reviewer 1 (Julia Rohrer)

Is the manuscript scientifically sound in its present form?

Yes

Are the interpretations and conclusions justified by the results?

Yes

Is the language acceptable?

No

Do you have any ethical concerns with this paper?

No

Have you any concerns about statistical analyses in this paper?

No

Recommendation?

Major revision is needed (please make suggestions in comments)

Comments to the Author(s)

This manuscript reports a close and successful replication of Newman and Cain's "tainted altruism" effect (2014). As you will see below, there is a couple of issues that I believe should be fixed prior to publication. For example, I found parts of manuscript (including parts of the structure, as well as some analyses) unnecessarily confusing. It may also be helpful to proofread the manuscript, some phrasings seemed a bit off to me (careful re-reading by all involved authors probably suffices). Lastly, I was not able to reproduce the numbers, it seems like the CSV provided does not really interface with the RMarkdown-File. This starts with the data provided having a different name than the data read in by the analysis script (I had enough patience to fix that on my end, but it's still suboptimal), but then variables seem to exist in the data the RMarkdown expects that don't exist under the same name in the data provided. I'm quite confident that the authors will be able to fix this without any issue (you just need to emulate what happens on the side of somebody who downloads the files and tries to execute the script).

Best regards,
Julia Rohrer

Bigger issues (which are still fairly small but go beyond typos):

- p. 3, "Substantially cited papers need and deserve independent replication. Until recently, it was often assumed that published results can be successfully replicated. However, as the famous Nosek et al (2015) reproducibility project indicates, only a modest fraction of published psychological research findings can be replicated.": I'm all in favor of brevity, but this summary of the replication crisis almost seems a bit comical. It may be just the phrasing though ("it was often assumed that published results can be successfully replicated" - really, was it? By whom? Did people assume everything could be replicated?); maybe you could give it another shock (maybe with more focus on "some people seemed genuinely surprised when the reproducibility project got dropped").

- p.6, "One additional question (not included in the original study) was asked in our experiment on an exploratory basis: "How deceptive do you think Mulberry is?""": When reading this, I was really wondering why you included the item. Starting on page 9, you provide an explanation which seems unnecessarily convoluted (and in any case, should have come much earlier). I fully understand what you are trying to say, but you could probably re-express it in 1-2 sentences that are much clearer (assuming you have fully worked out what you are trying to say here).

- p. 6, "providing us with a representative sample of people residing in the US": I find the notion that you could achieve an actually representative sample of people residing in the US through Prolific extremely implausible. The explanation behind that is a bit more complicated (it involves endogeneous selection bias) but not important for the purpose of the study. You can easily adjust your language to be accurate, i.e., talk about a sample representative of the US population with respects to age, sex, and ethnicity.

- p.7, averaging of questions: The original study probably did not do that either, but I'd really like to see Cronbach's alpha (or some other metric of item inter-correlation) every time you average things. You can easily include that information in the methods section. Now that I think about it: Some of the technical information when summarizing the original study may be better placed in the Methods section. That way, readers first get an overview of the design and findings; followed by all the nitty-gritty details.

- Figures: Unless I missed something, you currently do not reference the Figures in the text. Also, you report numbers in text that exactly correspond to values in the Figures, right? That seems a bit redundant (and cognitively unergonomic - lists of numbers in text are really not the optimal way to present information). The Figure caption also seems a bit overly specific for my taste (assuming the reader first reads the abstract and then skims the figure, they won't quite understand what is going on here),

- p. 8, "As to the question of how altruistically Mulberry acted, the average values were Charity = 3.98 (SE = 0.22), Advertising = 4.96 (SE = 0.18), Charity with Counterfactual = 5.29 (SE = 0.24), and Advertising with Counterfactual = 3.34 (SE = 0.19). The same cross-over interaction was evident as with the liking questions.": I'd really prefer to also have this in a figure. You could simply turn Figure 1 into a 4-panel figure and save quite a few words.

- p. 10, it seems quite interesting that for deceptiveness, there is no reversed pattern in the counterfactual condition. I was not sure whether you did not want to interpret/discuss this slightly different pattern (which could be justified, but then it should be explicitly stated); it does seem relevant for your previous reasoning about why to include deceptiveness in the first place (although, as stated above, I think this needs to be clarified to really make sense to readers)

- p. 10, "We compared the basic effect of tainted altruism, as defined in the results section, across groups of participants from several different demographic groups. The largest difference between genders appears in the male and female liking ratings of the subject in the Charity condition without counterfactual information ($M_{\text{male}} = 5.82$ and $M_{\text{female}} = 4.84$), a significant difference ($t(114) = 2.56, p = 0.005$). This indicates that female participants were significantly more likely than males to rate Mulberry less favorably when he donated to charity while gaining a profit.": I really don't understand what you did here and why. You report a comparison of two mean scores (liking ratings in the charity condition without counterfactual by gender). But, you are interested in moderating effects. Those do not concern simple differences between mean scores, but differences between effects, which should be tested accordingly. Also, it would be once again easier to report this with the help of figures. If you are really just interested in moderation of the tainted altruism effect, these would be very simple (x-axis: advertising condition, charity condition; separate lines for the two groups). Also, now that I think about it: Figure 1 is not quite fully aligned with what you are interested in - your central contrast is advertising vs charity, not counterfactual absent vs. counterfactual present. You might want to consider either re-arranging the plot to highlight the central contrast; or maybe add some labels in the plot so that readers see what the tainted altruism effect actually refers to.

- p. 11, "We also averaged the z-scores of political ideology and party questions from the demographic survey to form a single political measure, dividing the participants into four groups based on their relative z-scores. The examination of the tainted altruism effect present as a function of political attitudes did not disclose any systematic and comprehensible result.": This is the first time political ideology and party questions are mentioned, please include them in the methods section. Also, if this is an aggregate of two questions, please report their inter-correlation. Apart from this, I don't see how you could possibly justify diving people into four groups instead of just doing a proper interaction analysis with a continuous moderator. Please implement the appropriate analysis!

- p. 11, Experiment 2: In contrast to Experiment 1, here the description of the experiment is quite underspecified. For example, it was entirely unclear to me (until I saw the figure much later in the experiment) that by the eleven binary hiring decisions, you meant contrast with 11 different alternative promoters. Also, somewhere (preferably in a methods section), you should provide the specific contrasts.

- p. 13, just to be on the safe side – you link to the same dataset and same analysis script twice. That is intentional, right?

- p. 13, “Specifically, a participant was excluded if they chose the alternative firm, switched to Daniel’s firm, and then switched back to the alternative firm again.”: That does not make any sense to me, not even upon second re-reading. I have to assume that the options were ordered by attractiveness, so that “switching” indicates inconsistent preferences. However, at no point in the manuscript do you inform the reader about this.

- p. 14, “We used the averaged liking rating in Experiment 2 to evaluate the impact of data exclusion on our results. Two-sample independent t-tests reveal that the liking ratings before and after data exclusion in the Charity ($t(396) = 1.16, p = 0.24$) and Corporation ($t(327) = 0.90, p = 0.37$) conditions were not significantly different from each other. This indicates that the data exclusion caused by failing either of the comprehension checks in the original study does not significantly affect the results obtained in our replication.”: I do not understand what you are doing here and why. First of all, the t-test you did is guaranteed to be mis-specified (the two “independent samples” involve one sample that is a subset of the second sample). Second, this is not how you test for robustness to analytic decisions. The question is not whether the two means change, the question is whether the difference in the mean changes. The obvious analysis for this is: simply do not exclude the participants, redo the analysis and report the central effect of interest. If it’s about the same magnitude, you’re good.

Smaller issues:

p. 3, “has been cited over 150 times in other papers and research”: that seems like a rather odd phrasing to me. Is it supposed to reflect that Google Scholar also indexes non-paper research? Google Scholar also indexes blog posts (which wouldn’t be covered by “papers and research”), so you might want to use some other phrasing (such as “cited over 150 times according to Google Scholar (June 14, 2021)).

p. 4, “Direct Replication of Tainted Altruism Effect”: I really feel like this needs a “the”. Maybe it works without “the” for native speakers, but to me it seems like unnecessary telegraphic style (which is, of course, common in science).

p. 4, Section on Experiment 1: I found this a bit confusing because it is, in some sense, simultaneously telling me what happened in the original study and what happened in your replication. It might be helpful to be very explicit in the sections where you refer to findings of the original study (“Mulberry lost, rather than gained...” could be changed to “In the original study, Mulberry lost...”; although the second one was the one that actually confused me: “This did indeed reverse the effect (see below).” which could be changed to: “In the original study, this did indeed reverse the effect.” – or are you actually foreshadowing your own results with the see below? That would be a very weird decision, in my opinion)

p. 6, “During recruiting process”: Here, I once again feel like a “the” is needed.

p. 6, “Specifically, Prolific US representative sample”: Yet another “the” seems to be missing.

Review form: Reviewer 2

Is the manuscript scientifically sound in its present form?

Yes

Are the interpretations and conclusions justified by the results?

Yes

Is the language acceptable?

No

Do you have any ethical concerns with this paper?

No

Have you any concerns about statistical analyses in this paper?

No

Recommendation?

Accept with minor revision (please list in comments)

Comments to the Author(s)

This paper is a direct replication of 2 studies from Newman & Cain's influential 'Tainted Altruism' paper. As a direct replication, it was relatively easy to review, though I had a few comments. I'd like to say at the outset that it is heartening to see some classic social psych results holding up to scrutiny and that I do think the paper should be published. Nevertheless, I think it could do with some work to polish the paper. The introduction is very sparse and doesn't bring the reader up to speed on the literature. Given that Newman and Cain was published in 2014, I think there is room to discuss how things have progressed. In addition, the section on the importance of replication was very brief, amounting to just a couple of rather vague sentences.

There are at least 2 recent review articles that I know of which have summarised the state of the literature in the field and you might want to check them out.

Berman, J. Z., & Silver, I. (2021). Prosocial behavior and reputation: When does doing good lead to looking good?. *Current Opinion in Psychology*.

Raihani, N., & Power, E. A. (2021). No Good Deed Goes Unpunished: the social costs of prosocial behaviour. *Evolutionary Human Sciences*

- I found the writing to be a little clunky in several places e.g. " why say "human beings" rather than "people"? "Among the social psychologists" rather than "among social psychologists". this comment pertains to the presentation of the results as well. I am not sure how you might address this but I offer the comment in the spirit of being helpful rather than critical.

- Why did you only replicate studies 2 and 3 and not study 1? This needs explaining.

- It is also a little weird that you only replicate 2 of the 3 studies and then present them 'back to front' as it were - why not present them in the same order?

- I also thought you could have done more in the introduction to convince the reader why a replication was an interesting thing to do here. Rather than saying this subject has 'sparked much discussion; which is a bit of a cop out, it would be nice to see a more involved introduction, bringing us up to speed on where this literature has got to (see my next comment) as well as making the case for why a replication is valuable at this stage.

- P3, L17: the use of the term 'high-minded' in my mind is unnecessarily ambiguous as it could entail the motive underpinning the action. It would be better to use a term 'ostensibly prosocial' or similar as what you are trying to say is that sometimes things that look good on the outside might not stem from prosocial motives.

- Why were the last few words in the abstract italicized?
- It wasn't clear to me how the sample size was arrived at. Maybe it is detailed in the pre-registration document but it would be good to see some justification in the text.
- What happened to the 6 people who failed the attention check - were their data removed and was this exclusion pre-registered if so?
- In the paper you refer to sex whereas in the data it is gender. Best not to conflate the two and I think you probably asked people for their gender not their sex.
- page 7 - you say the effect was surprisingly large but would be good to give some justification, perhaps based on the effect size in the original study
- the rationale for the deceptiveness question does not appear until after the result is presented - it would be good to have some a priori justification for its inclusion, ideally in the introduction to the study.
- I wasn't super convinced by the 2-way interaction between gender and the counterfactual charity condition - I recognise that you also hedged this finding as well but I would query the validity of even running the analysis. Did you have a prior expectation of detecting an interaction here?

Decision letter (RSOS-211152.R0)

Dear Dr Pashler

The Editors assigned to your paper RSOS-211152 "Tainted Altruism: A Successful Pre-Registered Replication" have now received comments from reviewers and would like you to revise the paper in accordance with the reviewer comments and any comments from the Editors. Please note this decision does not guarantee eventual acceptance.

Please submit your revised manuscript and required files (see below) no later than 21 days from today's (ie 02-Sep-2021) date. Note: the ScholarOne system will 'lock' if submission of the revision is attempted 21 or more days after the deadline. If you do not think you will be able to meet this deadline please contact the editorial office immediately.

Please note article processing charges apply to papers accepted for publication in Royal Society Open Science (<https://royalsocietypublishing.org/rsos/charges>). Charges will also apply to papers transferred to the journal from other Royal Society Publishing journals, as well as papers submitted as part of our collaboration with the Royal Society of Chemistry

(<https://royalsocietypublishing.org/rsos/chemistry>). Fee waivers are available but must be requested when you submit your revision (<https://royalsocietypublishing.org/rsos/waivers>).

on behalf of Dr Giorgia Silani (Associate Editor) and Essi Viding (Subject Editor)
openscience@royalsociety.org

Associate Editor Comments to Author (Dr Giorgia Silani):

Associate Editor: 1

Comments to the Author:

Both reviewers consider your work potentially interesting and relevant but agree on the fact that the manuscript is still in a preliminary state, in terms of clarity and readability. Several points listed below need to be addressed before further consideration.

Reviewer comments to Author:

Reviewer: 1

Comments to the Author(s)

This manuscript reports a close and successful replication of Newman and Cain's "tainted altruism" effect (2014). As you will see below, there is a couple of issues that I believe should be fixed prior to publication. For example, I found parts of manuscript (including parts of the structure, as well as some analyses) unnecessarily confusing. It may also be helpful to proofread the manuscript, some phrasings seemed a bit off to me (careful re-reading by all involved authors probably suffices). Lastly, I was not able to reproduce the numbers, it seems like the CSV provided does not really interface with the RMarkdown-File. This starts with the data provided having a different name than the data read in by the analysis script (I had enough patience to fix that on my end, but it's still suboptimal), but then variables seem to exist in the data the RMarkdown expects that don't exist under the same name in the data provided. I'm quite confident that the authors will be able to fix this without any issue (you just need to emulate what happens on the side of somebody who downloads the files and tries to execute the script).

Best regards,
Julia Rohrer

Bigger issues (which are still fairly small but go beyond typos):

- p. 3, "Substantially cited papers need and deserve independent replication. Until recently, it was often assumed that published results can be successfully replicated. However, as the famous Nosek et al (2015) reproducibility project indicates, only a modest fraction of published psychological research findings can be replicated.": I'm all in favor of brevity, but this summary of the replication crisis almost seems a bit comical. It may be just the phrasing though ("it was often assumed that published results can be successfully replicated" - really, was it? By whom? Did people assume everything could be replicated?); maybe you could give it another shock (maybe with more focus on "some people seemed genuinely surprised when the reproducibility project got dropped").

- p.6, "One additional question (not included in the original study) was asked in our experiment on an exploratory basis: "How deceptive do you think Mulberry is?": When reading this, I was really wondering why you included the item. Starting on page 9, you provide an explanation which seems unnecessarily convoluted (and in any case, should have come much earlier). I fully understand what you are trying to say, but you could probably re-express it in 1-2 sentences that are much clearer (assuming you have fully worked out what you are trying to say here).

- p. 6, "providing us with a representative sample of people residing in the US": I find the notion that you could achieve an actually representative sample of people residing in the US through Prolific extremely implausible. The explanation behind that is a bit more complicated (it involves endogenous selection bias) but not important for the purpose of the study. You can easily adjust your language to be accurate, i.e., talk about a sample representative of the US population with respects to age, sex, and ethnicity.

- p.7, averaging of questions: The original study probably did not do that either, but I'd really like to see Cronbach's alpha (or some other metric of item inter-correlation) every time you average things. You can easily include that information in the methods section. Now that I think about it: Some of the technical information when summarizing the original study may be better placed in the Methods section. That way, readers first get an overview of the design and findings; followed by all the nitty-gritty details.

- Figures: Unless I missed something, you currently do not reference the Figures in the text. Also, you report numbers in text that exactly correspond to values in the Figures, right? That seems a bit redundant (and cognitively unergonomic - lists of numbers in text are really not the optimal way to present information). The Figure caption also seems a bit overly specific for my taste (assuming the reader first reads the abstract and then skims the figure, they won't quite understand what is going on here),

- p. 8, "As to the question of how altruistically Mulberry acted, the average values were Charity = 3.98 (SE = 0.22), Advertising = 4.96 (SE = 0.18), Charity with Counterfactual = 5.29 (SE = 0.24), and Advertising with Counterfactual = 3.34 (SE = 0.19). The same cross-over interaction was evident as with the liking questions.": I'd really prefer to also have this in a figure. You could simply turn Figure 1 into a 4-panel figure and save quite a few words.

- p. 10, it seems quite interesting that for deceptiveness, there is no reversed pattern in the counterfactual condition. I was not sure whether you did not want to interpret/discuss this slightly different pattern (which could be justified, but then it should be explicitly stated); it does seem relevant for your previous reasoning about why to include deceptiveness in the first place (although, as stated above, I think this needs to be clarified to really make sense to readers)

- p. 10, "We compared the basic effect of tainted altruism, as defined in the results section, across groups of participants from several different demographic groups. The largest difference between genders appears in the male and female liking ratings of the subject in the Charity condition without counterfactual information ($M_{\text{male}} = 5.82$ and $M_{\text{female}} = 4.84$), a significant difference ($t(114) = 2.56, p = 0.005$). This indicates that female participants were significantly more likely than males to rate Mulberry less favorably when he donated to charity while gaining a profit.": I really don't understand what you did here and why. You report a comparison of two mean scores (liking ratings in the charity condition without counterfactual by gender). But, you are interested in moderating effects. Those do not concern simple differences between mean scores, but differences between effects, which should be tested accordingly. Also, it would be once again easier to report this with the help of figures. If you are really just interested in moderation of the tainted altruism effect, these would be very simple (x-axis: advertising condition, charity condition; separate lines for the two groups). Also, now that I think about it: Figure 1 is not quite

fully aligned with what you are interested in – your central contrast is advertising vs charity, not counterfactual absent vs. counterfactual present. You might want to consider either re-arranging the plot to highlight the central contrast; or maybe add some labels in the plot so that readers see what the tainted altruism effect actually refers to.

- p. 11, “We also averaged the z-scores of political ideology and party questions from the demographic survey to form a single political measure, dividing the participants into four groups based on their relative z-scores. The examination of the tainted altruism effect present as a function of political attitudes did not disclose any systematic and comprehensible result.”: This is the first time political ideology and party questions are mentioned, please include them in the methods section. Also, if this is an aggregate of two questions, please report their inter-correlation. Apart from this, I don’t see how you could possibly justify diving people into four groups instead of just doing a proper interaction analysis with a continuous moderator. Please implement the appropriate analysis!

- p. 11, Experiment 2: In contrast to Experiment 1, here the description of the experiment is quite underspecified. For example, it was entirely unclear to me (until I saw the figure much later in the experiment) that by the eleven binary hiring decisions, you meant contrast with 11 different alternative promoters. Also, somewhere (preferably in a methods section), you should provide the specific contrasts.

- p. 13, just to be on the safe side – you link to the same dataset and same analysis script twice. That is intentional, right?

- p. 13, “Specifically, a participant was excluded if they chose the alternative firm, switched to Daniel’s firm, and then switched back to the alternative firm again.”: That does not make any sense to me, not even upon second re-reading. I have to assume that the options were ordered by attractiveness, so that “switching” indicates inconsistent preferences. However, at no point in the manuscript do you inform the reader about this.

- p. 14, “We used the averaged liking rating in Experiment 2 to evaluate the impact of data exclusion on our results. Two-sample independent t-tests reveal that the liking ratings before and after data exclusion in the Charity ($t(396) = 1.16, p = 0.24$) and Corporation ($t(327) = 0.90, p = 0.37$) conditions were not significantly different from each other. This indicates that the data exclusion caused by failing either of the comprehension checks in the original study does not significantly affect the results obtained in our replication.”: I do not understand what you are doing here and why. First of all, the t-test you did is guaranteed to be mis-specified (the two “independent samples” involve one sample that is a subset of the second sample). Second, this is not how you test for robustness to analytic decisions. The question is not whether the two means change, the question is whether the difference in the mean changes. The obvious analysis for this is: simply do not exclude the participants, redo the analysis and report the central effect of interest. If it’s about the same magnitude, you’re good.

Smaller issues:

p. 3, “has been cited over 150 times in other papers and research”: that seems like a rather odd phrasing to me. Is it supposed to reflect that Google Scholar also indexes non-paper research? Google Scholar also indexes blog posts (which wouldn’t be covered by “papers and research”), so you might want to use some other phrasing (such as “cited over 150 times according to Google Scholar (June 14, 2021)).

p. 4, “Direct Replication of Tainted Altruism Effect”: I really feel like this needs a “the”. Maybe it works without “the” for native speakers, but to me it seems like unnecessary telegraphic style (which is, of course, common in science).

p. 4, Section on Experiment 1: I found this a bit confusing because it is, in some sense, simultaneously telling me what happened in the original study and what happened in your replication. It might be helpful to be very explicit in the sections where you refer to findings of the original study ("Mulberry lost, rather than gained..." could be changed to "In the original study, Mulberry lost..."; although the second one was the one that actually confused me: "This did indeed reverse the effect (see below)." which could be changed to: "In the original study, this did indeed reverse the effect." - or are you actually foreshadowing your own results with the see below? That would be a very weird decision, in my opinion)

p. 6, "During recruiting process": Here, I once again feel like a "the" is needed.

p. 6, "Specifically, Prolific US representative sample": Yet another "the" seems to be missing.

Reviewer: 2

Comments to the Author(s)

This paper is a direct replication of 2 studies from Newman & Cain's influential 'Tainted Altruism' paper. As a direct replication, it was relatively easy to review, though I had a few comments. I'd like to say at the outset that it is heartening to see some classic social psych results holding up to scrutiny and that I do think the paper should be published. Nevertheless, I think it could do with some work to polish the paper. The introduction is very sparse and doesn't bring the reader up to speed on the literature. Given that Newman and Cain was published in 2014, I think there is room to discuss how things have progressed. In addition, the section on the importance of replication was very brief, amounting to just a couple of rather vague sentences.

There are at least 2 recent review articles that I know of which have summarised the state of the literature in the field and you might want to check them out.

Berman, J. Z., & Silver, I. (2021). Prosocial behavior and reputation: When does doing good lead to looking good?. *Current Opinion in Psychology*.

Raihani, N., & Power, E. A. (2021). No Good Deed Goes Unpunished: the social costs of prosocial behaviour. *Evolutionary Human Sciences*

- I found the writing to be a little clunky in several places e.g. " why say "human beings" rather than "people"? "Among the social psychologists" rather than "among social psychologists". this comment pertains to the presentation of the results as well. I am not sure how you might address this but I offer the comment in the spirit of being helpful rather than critical.

- Why did you only replicate studies 2 and 3 and not study 1? This needs explaining.

- It is also a little weird that you only replicate 2 of the 3 studies and then present them 'back to front' as it were - why not present them in the same order?

- I also thought you could have done more in the introduction to convince the reader why a replication was an interesting thing to do here. Rather than saying this subject has 'sparked much discussion; which is a bit of a cop out, it would be nice to see a more involved introduction, bringing us up to speed on where this literature has got to (see my next comment) as well as making the case for why a replication is valuable at this stage.

- P3, L17: the use of the term 'high-minded' in my mind is unnecessarily ambiguous as it could entail the motive underpinning the action. It would be better to use a term 'ostensibly prosocial'

or similar as what you are trying to say is that sometimes things that look good on the outside might not stem from prosocial motives.

- Why were the last few words in the abstract italicized?
- It wasn't clear to me how the sample size was arrived at. Maybe it is detailed in the pre-registration document but it would be good to see some justification in the text.
- What happened to the 6 people who failed the attention check - were their data removed and was this exclusion pre-registered if so?
- In the paper you refer to sex whereas in the data it is gender. Best not to conflate the two and I think you probably asked people for their gender not their sex.
- page 7 - you say the effect was surprisingly large but would be good to give some justification, perhaps based on the effect size in the original study
- the rationale for the deceptiveness question does not appear until after the result is presented - it would be good to have some a priori justification for its inclusion, ideally in the introduction to the study.
- I wasn't super convinced by the 2-way interaction between gender and the counterfactual charity condition - I recognise that you also hedged this finding as well but I would query the validity of even running the analysis. Did you have a prior expectation of detecting an interaction here?

===PREPARING YOUR MANUSCRIPT===

===PREPARING YOUR REVISION IN SCHOLARONE===

Author's Response to Decision Letter for (RSOS-211152.R0)

See Appendix A.

RSOS-211152.R1 (Revision)

Review form: Reviewer 1 (Julia Rohrer)

Is the manuscript scientifically sound in its present form?

Yes

Are the interpretations and conclusions justified by the results?

Yes

Is the language acceptable?

Yes

Do you have any ethical concerns with this paper?

No

Have you any concerns about statistical analyses in this paper?

No

Recommendation?

Accept as is

Comments to the Author(s)

I would like to thank the authors for addressing all of my comments in a satisfying manner.

There is only a tiny concern that remains: I still can't get the RMarkdown to run. If I follow the Figshare link, I download a csv file labelled "Pashler replication of Newman and Cain for Figshare 2021.csv". However, If I use the provided RMarkdown Script, this calls for a file named "TaintedAltruisticNumeric.csv." This is not just a difference in the naming of the csv file; the script does not run with the data provided online because numerous variables are missing (or maybe they are read erroneously, because the csv provided has a different structure).

Given that the authors already went through the hassle of making their analyses reproducible with an RMarkdown-Script, I think it would be a shame if the correct materials weren't shared. The response to the reviewers states that now an xlsx file is used and something was done to make the structure more transparent and readable. However, the figshare link still leads to the csv and an RMarkdown file that calls for a different csv -- maybe the corresponding links weren't updated in the manuscript?

Review form: Reviewer 2

Is the manuscript scientifically sound in its present form?

Yes

Are the interpretations and conclusions justified by the results?

Yes

Is the language acceptable?

Yes

Do you have any ethical concerns with this paper?

No

Have you any concerns about statistical analyses in this paper?

No

Recommendation?

Accept as is

Comments to the Author(s)

This is a semi-responsive revision. I feel the lit review is still a little bit superficial but it's not my paper and the authors can write as much or as little as they want about both tainted altruism and the replication crisis. The abstract ends a little abruptly and could do with a general summary sentence. I also think overlaying raw data on figures 1 and 2 would be helpful.

Decision letter (RSOS-211152.R1)

Dear Dr Pashler,

It is a pleasure to accept your manuscript entitled "Tainted Altruism: A Successful Pre-Registered Replication" in its current form for publication in Royal Society Open Science. The comments of the reviewer(s) who reviewed your manuscript are included at the foot of this letter.

The proof of your paper will be available for review using the Royal Society online proofing system and you will receive details of how to access this in the near future from our production office (openscience_proofs@royalsociety.org). We aim to maintain rapid times to publication after acceptance of your manuscript and we would ask you to please contact both the production office

and editorial office if you are likely to be away from e-mail contact to minimise delays to publication. If you are going to be away, please nominate a co-author (if available) to manage the proofing process, and ensure they are copied into your email to the journal.

on behalf of Dr Giorgia Silani (Associate Editor) and Essi Viding (Subject Editor)
 openscience@royalsociety.org

Associate Editor Comments to Author (Dr Giorgia Silani):

Associate Editor: 1

Comments to the Author:

Dear authors, both reviewers have expressed satisfaction with your replies. Please be aware of an additional point raised by reviewer 2 that you may address. The comment states:

"There is only a tiny concern that remains: I still can't get the RMarkdown to run. If I follow the Figshare link, I download a csv file labelled "Pashler replication of Newman and Cain for Figshare 2021.csv". However, If I use the provided RMarkdown Script, this calls for a file named "TaintedAltruisticNumeric.csv." This is not just a difference in the naming of the csv file; the script does not run with the data provided online because numerous variables are missing (or maybe they are read erroneously, because the csv provided has a different structure).

Given that the authors already went through the hassle of making their analyses reproducible with an RMarkdown-Script, I think it would be a shame if the correct materials weren't shared. The response to the reviewers states that now an xlsx file is used and something was done to make the structure more transparent and readable. However, the figshare link still leads to the csv and an RMarkdown file that calls for a different csv -- maybe the corresponding links weren't updated in the manuscript? "

Reviewer comments to Author:

Reviewer: 2

Comments to the Author(s)

This is a semi-responsive revision. I feel the lit review is still a little bit superficial but it's not my paper and the authors can write as much or as little as they want about both tainted altruism and the replication crisis. The abstract ends a little abruptly and could do with a general summary sentence. I also think overlaying raw data on figures 1 and 2 would be helpful.

Reviewer: 1

Comments to the Author(s)

I would like to thank the authors for addressing all of my comments in a satisfying manner.

There is only a tiny concern that remains: I still can't get the RMarkdown to run. If I follow the Figshare link, I download a csv file labelled "Pashler replication of Newman and Cain for Figshare 2021.csv". However, If I use the provided RMarkdown Script, this calls for a file named "TaintedAltruisticNumeric.csv." This is not just a difference in the naming of the csv file; the script does not run with the data provided online because numerous variables are missing (or maybe they are read erroneously, because the csv provided has a different structure).

Given that the authors already went through the hassle of making their analyses reproducible with an RMarkdown-Script, I think it would be a shame if the correct materials weren't shared. The response to the reviewers states that now an xlsx file is used and something was done to make the structure more transparent and readable. However, the figshare link still leads to the csv and an RMarkdown file that calls for a different csv -- maybe the corresponding links weren't updated in the manuscript?

Appendix A

Dear Drs. Viding and Silani.

Thank you very much for the opportunity to revise manuscript RSOS-211152.

Editors and reviewers have provided extremely useful comments. We have tried to respond to all of them below, generally by describing a change we have made in reaction to the comment.

We attach several documents. One is the Action letter (and reviews) with our responses shown **in blue ink**.

We also attach a revision with changes shown **in green ink** (as well as a clean copy with everything in black ink). These are the documents whose names end in `_green` and `_clean`, respectively

We apologize that the revision process took us more time than anticipated, and we needed to repeatedly request extensions from your end. We appreciate your patience.

Hal Pashler, corresponding author

-----***

Dear Dr Pashler

The Editors assigned to your paper RSOS-211152 "Tainted Altruism: A Successful Pre-Registered Replication" have now received comments from reviewers and would like you to revise the paper in accordance with the reviewer comments and any comments from the Editors. Please note this decision does not guarantee eventual acceptance.

Please submit your revised manuscript and required files (see below) no later than 21 days from today's (ie 02-Sep-2021) date. Note: the ScholarOne system will 'lock' if submission of the

revision is attempted 21 or more days after the deadline. If you do not think you will be able to meet this deadline please contact the editorial office immediately.

on behalf of Dr Giorgia Silani (Associate Editor) and Essi Viding (Subject Editor)
openscience@royalsociety.org

Associate Editor Comments to Author (Dr Giorgia Silani):

Associate Editor: 1

Comments to the Author:

Both reviewers consider your work potentially interesting and relevant but agree on the fact that the manuscript is still in a preliminary state, in terms of clarity and readability. Several points listed below need to be addressed before further consideration.

Reviewer comments to Author:

Reviewer: 1

Comments to the Author(s)

This manuscript reports a close and successful replication of Newman and Cain's "tainted altruism" effect (2014). As you will see below, there is a couple of issues that I believe should be fixed prior to publication. For example, I found parts of manuscript (including parts of the structure, as well as some analyses) unnecessarily confusing. It may also be helpful to proofread the manuscript, some phrasings seemed a bit off to me (careful re-reading by all involved authors probably suffices). Lastly, I was not able to reproduce the numbers, it seems like the CSV provided does not really interface with the RMarkdown-File. This starts with the data provided having a different name than the data read in by the analysis script (I had enough patience to fix that on my end, but it's still suboptimal), but then variables seem to exist in the data the RMarkdown expects that don't exist under the same name in the data provided. I'm

quite confident that the authors will be able to fix this without any issue (you just need to emulate what happens on the side of somebody who downloads the files and tries to execute the script).

Response: The problem that Reviewer 1 experienced might have occurred because symbols within cells in .csv files could not be correctly processed by RMarkdown. We use an .xlsx file to circumvent this problem and divide the data by conditions to make its structure more transparent and readable. We expect readers should now be able to analyze our data with RMarkdown.

Best regards,
Julia Rohrer

In addition to the many reviewer-requested changes listed here, we also changed the description of the conditions in the first experiment to be more concrete and clearer, with more verbatim materials reducing the demand on the reader.

Bigger issues (which are still fairly small but go beyond typos):

- p. 3, “Substantially cited papers need and deserve independent replication. Until recently, it was often assumed that published results can be successfully replicated. However, as the famous Nosek et al (2015) reproducibility project indicates, only a modest fraction of published psychological research findings can be replicated.”: I’m all in favor of brevity, but this summary of the replication crisis almost seems a bit comical.

It may be just the phrasing though (“it was often assumed that published results can be successfully replicated” – really, was it? By whom? Did people assume everything could be replicated?);

It seems that we may have differing recollections and experiences in this regard. The member of our team who has been following the replication crisis most closely since its inception (author HP) makes the following comment: “It may have been naive of me, but I *was* truly appalled and shocked to learn that large portions of major fields of experimental psychology were strewn with imaginary results. I had assumed such things happened occasionally and not all the time. Moreover, during the time we were all anticipating the results of the Nosek et al. RPP project, I recall specifically asking colleagues and friends to guess what proportion of results would fail to replicate, and the estimates I heard were almost all well below the actual proportion that came out of the study. My own estimate was 15% and I told people that were it to turn out that high it would be viewed as scandalously high, a black mark on the field. So I know the naive onlookers we describe in our paper really existed, because I was one!”

Nonetheless, given the referee’s reactions we have toned our language down a bit on this point.

maybe you could give it another shock (maybe with more focus on “some people seemed genuinely surprised when the reproducibility project got dropped”).

- p.6, “One additional question (not included in the original study) was asked in our experiment on an exploratory basis: “How deceptive do you think Mulberry is?””: When reading this, I was really wondering why you included the item. Starting on page 9, you provide an explanation which seems unnecessarily convoluted (and in any case, should have come much earlier). I fully understand what you are trying to say, but you could probably re-express it in 1-2 sentences that are much clearer (assuming you have fully worked out what you are trying to say here).

We agree the explanation was convoluted and we have shortened it a lot. We see why the reviewer found it jarring that no explanation for inserting the deceptiveness question was mentioned when it was first brought up, so we now say “for reasons that will be described below”. (It is, in our judgment, too early in the paper to go into this in detail yet.)

p. 6, “providing us with a representative sample of people residing in the US”: I find the notion that you could achieve an actually representative sample of people residing in the US through Prolific extremely implausible. The explanation behind that is a bit more complicated (it involves endogeneous selection bias) but not important for the purpose of the study. You can easily adjust your language to be accurate, i.e., talk about a sample representative of the US population with respects to age, sex, and ethnicity.

The referee makes a very interesting point and we share her skepticism that any dataset composed of volunteers can ever be fully representative of a population (at least it seems sure to be unrepresentative in regard to some trait like propensity to volunteer for studies, which is surely not orthogonal to agreeability). The referee’s point thus probably applies far beyond Prolific samples, e.g., it may extends to famous large scale research initiatives that purport to study representative samples. In any case, our text has been modified in line with the reviewer’s suggestion. A slightly trickier issue was how to handle this in the Abstract, where we had also bragged on the “representative” sample. We changed that language so it now reads “using a larger sample (n=501) intended to be fairly representative of the US population” which seems accurate since Prolific makes it clear that they intend exactly that (and ‘intended’ implicitly concedes the aspirational nature of this 😊). While we agree that the virtues of Prolific shouldn’t be overstated, we don’t think it would make sense to to slam on this valuable new resource nor are we the people to get into thorny issues in sampling theory that are beyond our ken (many statisticians have kicked this problem around along with political scientists, and we doubt we have much to add or that our readers would be interested). Our sample is what it is, and we think at a minimum, the sample is more likely than a college convenience sample to yield accurate out-of-sample projections even if it falls short of ideal representativeness.)

- p.7, averaging of questions: The original study probably did not do that either, but I'd really like to see Cronbach's alpha (or some other metric of item inter-correlation) every time you average things.

You can easily include that information in the methods section. Now that I think about it: Some of the technical information when summarizing the original study may be better placed in the Methods section. That way, readers first get an overview of the design and findings; followed by all the nitty-gritty details.

The reviewer's point is well taken and we have now reported Cronbach's alpha wherever we aggregate. The value was .899 for the liking questions in Experiment 1, .927 for the morality questions in Experiment 2, and .844 for the benefit questions in Experiment 2. We chose to put all these figures in the Results sections.

- Figures: Unless I missed something, you currently do not reference the Figures in the text.

Response: Thanks for noting that, it has been fixed (see p. 9 and other spots).

Response: Also, you report numbers in text that exactly correspond to values in the Figures, right? That seems a bit redundant (and cognitively unergonomic – lists of numbers in text are really not the optimal way to present information).

We respectfully disagree that presenting a handful of numbers which are also derivable from a figure (if you have the right software) is a bad idea for ergonomic reasons. We think this sort of minor redundancy is harmless and promotes convenience and comprehension. However, we have reduced redundancy a bit in the revision.

The Figure caption also seems a bit overly specific for my taste (assuming the reader first reads the abstract and then skims the figure, they won't quite understand what is going on here),

We agree and have shortened all the captions.

- p. 8, "As to the question of how altruistically Mulberry acted, the average values were Charity = 3.98 (SE = 0.22), Advertising = 4.96 (SE = 0.18), Charity with Counterfactual = 5.29 (SE = 0.24), and Advertising with Counterfactual = 3.34 (SE = 0.19). The same cross-over interaction was evident as with the liking questions.": I'd really prefer to also have this in a figure. You could simply turn Figure 1 into a 4-panel figure and save quite a few words.

We have added an additional figure although we could not create the suggested 4-panel figure because the original authors didn't provide everything needed for that.

- p. 10, it seems quite interesting that for deceptiveness, there is no reversed pattern in the counterfactual condition. I was not sure whether you did not want to interpret/discuss this slightly different pattern (which could be justified, but then it should be explicitly stated); it does

seem relevant for your previous reasoning about why to include deceptiveness in the first place (although, as stated above, I think this needs to be clarified to really make sense to readers)

The counterfactual information did not produce a reversal but it seemed to moderate (reduce) the effect a bit. In our view, if you actually spell out precise theories of the effect, you find that they don't make predictions of reversal, just moderation. Thus, we don't spend much time on this in the manuscript. (We also doubt that the crossovers have the strong theoretical implications the original authors assumed in the first place, but that's a more complicated discussion.) Anyway, all our data are now public for anyone who thinks there is theoretical leverage to be had from more detailed comparisons of various kinds.

- p. 10, "We compared the basic effect of tainted altruism, as defined in the results section, across groups of participants from several different demographic groups. The largest difference between genders appears in the male and female liking ratings of the subject in the Charity condition without counterfactual information ($M_{\text{male}} = 5.82$ and $M_{\text{female}} = 4.84$), a significant difference ($t(114) = 2.56, p = 0.005$). This indicates that female participants were significantly more likely than males to rate Mulberry less favorably when he donated to charity while gaining a profit." I really don't understand what you did here and why. You report a comparison of two mean scores (liking ratings in the charity condition without counterfactual by gender). But, you are interested in moderating effects. Those do not concern simple differences between mean scores, but differences between effects, which should be tested accordingly. Also, it would be once again easier to report this with the help of figures. If you are really just interested in moderation of the tainted altruism effect, these would be very simple (x-axis: advertising condition, charity condition; separate lines for the two groups). Also, now that I think about it: Figure 1 is not quite fully aligned with what you are interested in – your central contrast is advertising vs charity, not counterfactual absent vs. counterfactual present. You might want to consider either re-arranging the plot to highlight the central contrast; or maybe add some labels in the plot so that readers see what the tainted altruism effect actually refers to.

We made the figures requested by the referee, breaking the data down by gender (see below). Doing so immediately convinced us that there is nothing in our results to warrant "teasing" the readers about a possible gender difference as we had done in the first version. So we took it out. We are embarrassed that we did not start with this data visualization as we often do. (In our opinion, it makes sense to mention glaringly obvious patterns observed post hoc in the data as "teasers" when they are strong enough, but this gender thing was really not strong to start with.)

- p. 11, “We also averaged the z-scores of political ideology and party questions from the demographic survey to form a single political measure, dividing the participants into four groups based on their relative z-scores. The examination of the tainted altruism effect present as a function of political attitudes did not disclose any systematic and comprehensible result.”: This is the first time political ideology and party questions are mentioned, please include them in the methods section. Also, if this is an aggregate of two questions, please report their inter-correlation. Apart from this, I don’t see how you could possibly justify dividing people into four groups instead of just doing a proper interaction analysis with a continuous moderator. Please implement the appropriate analysis!

Here too, we decided on reflection that it makes little sense to drag the reader through these completely null effects which we had looked at in the first place out of mere curiosity. Since neither of the reviewers seemed to find much value in mentioning these analyses either, we are happy to have removed them.

- p. 11, Experiment 2: In contrast to Experiment 1, here the description of the experiment is quite underspecified. For example, it was entirely unclear to me (until I saw the figure much later in the experiment) that by the eleven binary hiring decisions, you meant contrast with 11 different alternative promoters.

We have added further details (see p. 13).

Also, somewhere (preferably in a methods section), you should provide the specific contrasts.

We now provide the full sequence of exact wordings of the choices and questions (see Supplementary Online Materials).

- p. 13, just to be on the safe side - you link to the same dataset and same analysis script twice. This is intentional, right?

Yes, this was intentional.

- p. 13, "Specifically, a participant was excluded if they chose the alternative firm, switched to Daniel's firm, and then switched back to the alternative firm again.": That does not make any sense to me, not even upon second re-reading. I have to assume that the options were ordered by attractiveness, so that "switching" indicates inconsistent preferences. However, at no point in the manuscript do you inform the reader about this.

This was indeed confusing. We reworded it more formally (p. 14) and briefly mentioned what we surmise to be the original authors' reason for imposing this constraint.

- p. 14, "We used the averaged liking rating in Experiment 2 to evaluate the impact of data exclusion on our results. Two-sample independent t-tests reveal that the liking ratings before and after data exclusion in the Charity ($t(396) = 1.16, p = 0.24$) and Corporation ($t(327) = 0.90, p = 0.37$) conditions were not significantly different from each other. This indicates that the data exclusion caused by failing either of the comprehension checks in the original study does not significantly affect the results obtained in our replication.": I do not understand what you are doing here and why. First of all, the t-test you did is guaranteed to be mis-specified (the two "independent samples" involve one sample that is a subset of the second sample). Second, this is not how you test for robustness to analytic decisions. The question is not whether the two means change, the question is whether the difference in the mean changes. The obvious analysis for this is: simply do not exclude the participants, redo the analysis and report the central effect of interest. If it's about the same magnitude, you're good.

The reviewer made a great point here. On reflection, we agree that the previous analysis was conceptually botched. We have now proceeded as the referee suggested, and calculated the morality and benefit ratings comparisons before any data were excluded. As described in the revised ms, before any data exclusion, the difference of morality ratings between both conditions was statistically significant and the difference of benefit ratings between both conditions was not statistically significant. These results all line up with that was found with the data exclusion so we now have a sensible grounds for dismissing concerns about the severe exclusion policy.

Smaller issues:

p. 3, "has been cited over 150 times in other papers and research": that seems like a rather odd phrasing to me. Is it supposed to reflect that Google Scholar also indexes non-paper research? Google Scholar also indexes blog posts (which wouldn't be covered by "papers and research"), so you might want to use some other phrasing (such as "cited over 150 times according to Google Scholar (June 14, 2021)).

We agree and adopted the suggested wording (see p. 3).

p. 4, "Direct Replication of Tainted Altruism Effect": I really feel like this needs a "the". Maybe it works without "the" for native speakers, but to me it seems like unnecessary telegraphic style (which is, of course, common in science).

We agree and have made the proposed change (title page).

p. 4, Section on Experiment 1: I found this a bit confusing because it is, in some sense, simultaneously telling me what happened in the original study and what happened in your replication. It might be helpful to be very explicit in the sections where you refer to findings of the original study ("Mulberry lost, rather than gained..." could be changed to "In the original study, Mulberry lost...";

Suggestion taken (p. 6).

...although the second one was the one that actually confused me: "This did indeed reverse the effect (see below)." which could be changed to: "In the original study, this did indeed reverse the effect."

Suggestion taken (p. 6).

– or are you actually foreshadowing your own results with the see below? That would be a very weird decision, in my opinion)

Agreed, it would be. :)

p. 6, "During recruiting process": Here, I once again feel like a "the" is needed.

Error fixed (p. 6).

p. 6, "Specifically, Prolific US representative sample": Yet another "the" seems to be missing.

Fixed (p. 6).

Reviewer: 2

Comments to the Author(s)

This paper is a direct replication of 2 studies from Newman & Cain's influential 'Tainted Altruism' paper. As a direct replication, it was relatively easy to review, though I had a few comments. I'd like to say at the outset that it is heartening to see some classic social psych results holding up to scrutiny and that I do think the paper should be published. Nevertheless, I think it could do with some work to polish the paper. The introduction is very sparse and doesn't bring the reader

up to speed on the literature. Given that Newman and Cain was published in 2014, I think there is room to discuss how things have progressed. In addition, the section on the importance of replication was very brief, amounting to just a couple of rather vague sentences.

We agree that our discussion of the importance of replication was too terse, and we have beefed it up slightly. On the other hand, we do not agree that it makes sense for us to insert more elaborate literature review--especially because we can now cite the two very recent review articles on the topic that the reviewer was kind enough to mention.

There are at least 2 recent review articles that I know of which have summarised the state of the literature in the field and you might want to check them out.

Berman, J. Z., & Silver, I. (2021). Prosocial behavior and reputation: When does doing good lead to looking good?. *Current Opinion in Psychology*.

Raihani, N., & Power, E. A. (2021). No Good Deed Goes Unpunished: the social costs of prosocial behaviour. *Evolutionary Human Sciences*

Again, we are grateful for these and have added citations to them.

- I found the writing to be a little clunky in several places e.g. " why say "human beings" rather than "people"?

We agree and we have replaced 'human beings' with 'people', as suggested.

"Among the social psychologists" rather than "among social psychologists". this comment pertains to the presentation of the results as well. I am not sure how you might address this but I offer the comment in the spirit of being helpful rather than critical.

We agree about this also and have made the proposed change.

- Why did you only replicate studies 2 and 3 and not study 1? This needs explaining.

We had several reasons, starting with the fact that Exp 1 with its focus on within- versus between-subject measurements struck us as somewhat secondary in importance and potentially distracting. We added a one-sentence description of our reasons (it would be complicated and boring for the reader for us to explain the rationale in greater detail, it seems to us).

- It is also a little weird that you only replicate 2 of the 3 studies and then present them 'back to front' as it were - why not present them in the same order?

Naturally, it was a conscious choice that we had discussed extensively among ourselves before writing the paper up. We felt that it makes more sense to readers when presented in that order.

This way we present the basic effect from Exp 3 first, a wrinkle on it with the counterfactual information next, and then the quantitative titration of the effect in Experiment 2 last.

- I also thought you could have done more in the introduction to convince the reader why a replication was an interesting thing to do here. Rather than saying this subject has 'sparked much discussion; which is a bit of a cop out, it would be nice to see a more involved introduction, bringing us up to speed on where this literature has got to (see my next comment) as well as making the case for why a replication is valuable at this stage.

As discussed above, we do not believe that this empirically focused paper would benefit from inclusion of any elaborate literature review and we do not believe that readers would expect to see it here. The referees seem to misunderstand our motivation slightly when they refer to "valuable at this stage", suggesting we must view these particular results as somehow especially crucial to the narrow field of which it is a part. We believe that all findings that have attained high profiles merit replication regardless of the results, not just a hand-picked few (see Phillips, Harris, and Wixted, 2020, for a statistical analysis of this built around plausible priors for true effect sizes.)

- - P3, L17: the use of the term 'high-minded' in my mind is unnecessarily ambiguous as it could entail the motive underpinning the action. It would be better to use a term 'ostensibly prosocial' or similar as what you are trying to say is that sometimes things that look good on the outside might not stem from prosocial motives.

This strikes us as an excellent suggestion and we have adopted it.

- Why were the last few words in the abstract italicized?

This was an error, and it has been fixed.

- It wasn't clear to me how the sample size was arrived at. Maybe it is detailed in the pre-registration document but it would be good to see some justification in the text.

The original samples were 92 and 145 in Expts 2 and 3, respectively, which generally yielded clearly significant results, not vague trends. We were pleased that we could afford to test substantially more people than the original, and also enough to go slightly beyond the level of 2.5 times the original study's n that is recommended by Uri Simonsohn in his thoughtful 2015 survey of this complex topic (Simonsohn, 2015, Small telescopes: Detectability and the evaluation of replication results. *Psychological science*, 26, 559-569.). So everything suggested going with the nice round number of 500 (exactly how we ended up with 501 rather than 500 is not interesting enough to discuss here.)

What happened to the 6 people who failed the attention check - were their data removed and was this exclusion pre-registered if so?

Their data were excluded as per the original study and the pre-registration.

- In the paper you refer to sex whereas in the data it is gender. Best not to conflate the two and I think you probably asked people for their gender not their sex.

We have now deleted the presentation of the iffy results bearing on gender, so most of the occurrences of these terms are now gone. In fact, though, we strongly suspect that in its initial recruitment Prolific *does* ask subjects questions for which it uses the term 'sex', since they name fields in their database with 'sex' rather than 'gender', so the premise of the question is not quite right. Nothing important hinges on any of this, as far as we can see, so we don't go into it in the manuscript.

- page 7 - you say the effect was surprisingly large but would be good to give some justification, perhaps based on the effect size in the original study

Our rationale for saying that is based on the rarity of confirmable effects of that size in the psychological literature in general, as e.g., discussed by Simmons, J. P., Nelson, L. D., & Simonsohn, U. (2013, January). Life after p-hacking. In *Meeting of the society for personality and social psychology, New Orleans, LA* (pp. 17-19) (not cited because we can't find anything citable). Meta-analyses looking at effect sizes in the published literature seem to us useless for deciding what effect sizes are truly rare, because these are so distorted by publication bias.

- the rationale for the deceptiveness question does not appear until after the result is presented
- it would be good to have some a priori justification for its inclusion, ideally in the introduction to the study.

Yes, this is a point both reviewers mentioned and we agree. We now mention that a rationale will be forthcoming and we have streamlined the description of that rationale and interpretation of the results.

- I wasn't super convinced by the 2-way interaction between gender and the counterfactual charity condition - I recognise that you also hedged this finding as well but I would query the validity of even running the analysis. Did you have a prior expectation of detecting an interaction here?

As discussed above, we now agree with the reviewer about this and we have deleted mention of this very weak trend.